

# Surprisingly high levels and activity contributions of oxygenated volatile organic compounds on the southeast of the Tibetan Plateau

Shuzheng Guo[1], Chunxiang Ye[2], Weili Lin[1], Yi Chen[1], Limin Zeng[2], Xuena Yu[2], Jinhui Cui[2], Chong Zhang[2], Jing Duan[3], Haobin Zhong[3], Rujin Huang[3], Xuguang Chi[4], Wei Nie[4], Aijun Ding[4]

[1]Key Laboratory of Ecology and Environment in Minority Areas, Minzu University of China, National Ethnic Affairs Commission, Beijing 100081, China

[2]College of Environmental Science and Technology, Peking University, Beijing 100871, China

[3]Key Laboratory of Aerosol Chemistry and Physics, Institute of Earth Environment, Chinese Academy of Sciences, Xi'an 710075, China

[4]National Observation and Research Station for Atmospheric Processes and Environmental Change in Yangtze River Delta, Nanjing 210046, China

*Correspondence:* c.ye@pku.edu.cn, linwl@muc.edu.cn

**Abstract.** Oxygenated volatile organic compounds (OVOCs) are reactive species and the primary precursors of free radicals; thus, OVOCs play important roles in tropospheric chemistry. @Tibet field campaigns 2021 discovered surprisingly high levels and activity contributions of OVOCs at Lulang, a site with high vegetation cover and strong solar ultraviolet radiation on the southeast of the Tibetan Plateau (TP). The 13 OVOCs detected accounted for 49% of the total VOCs (TVOCs; average level of $11.7 \pm 4.4$ ppb), and the levels of these OVOCs exhibited typical diurnal variation, with high values in the daytime and a peak at approximately 12:00. OVOCs contributed 65% and 63% to VOC-$k_{OH}$ and the ozone formation potential, respectively, and thus had a strong influence on atmospheric chemical processes. Two independent methods were used to determine the contributions of various sources and revealed consistent conclusions regarding the importance of biological sources there. The source apportionment results obtained through positive matrix factorization indicated that sunlight-impacted and direct plant emission sources both related to plant sources contributed 47% of the TVOCs and 65% of the OVOCs. OVOC source fitting through the photochemical age parameterization method also indicated that biogenic sources made the largest contribution (67%) to OVOCs and revealed a clear peak at noon. In addition, biomass burning sources were found to be closely related to the VOC background because biomass burning is highly prevalent across the whole TP; these sources made the second greatest contribution (33%) to the TVOCs and contributed more than 23% of OVOCs.

## 1 Introduction

In the troposphere, numerous volatile organic compounds (VOCs) and their oxidative products [i.e., oxygenated VOCs (OVOCs)] are reactive species in hydroxyl radical oxidation or photolysis (Atkinson, 2000). In the process of VOCs oxidation, a series of secondary photochemical products are produced and accumulate. For instance, ozone, organic nitrates, and secondary organic aerosols (SOAs) are major products of VOC oxidation (Derwent et al., 1996; Zhang et al., 2014a; Derwent et al., 2010). Some reactive VOCs are toxic to the human body or ecological systems (Galloway et al., 1982; Lyu et al., 2020). Therefore, VOCs impact human health, air quality, and climate change, whether directly or indirectly.

VOCs mainly originate from natural cycles, anthropogenic activity, and photochemical formation (Atkinson and Arey, 2003; Guenther et al., 2000; Mozaffar and Zhang, 2020). Anthropogenic VOCs contribute heavily to the atmospheric levels of the major air pollutants $O_3$ and $PM_{2.5}$, especially in high-pollution regions. Biogenic VOCs emitted from forested areas and city trees are also a major source of VOCs in terms of both the carbon mass inventory and overall reactivity to hydroxyl radicals. In addition, the oxidation processes of biogenic VOCs characterize the efficiency of OH recycling and the chemical formation



of $O_3$. The atmospheric fate of VOCs, especially accumulated OVOCs, has thus been a focus of research.

The Tibetan Plateau (TP), with an average altitude of over 4000 m and a total area of approximately 2.5 million $km^2$, is located in the middle and lower troposphere and experiences strong solar radiation. The TP has a sparse population and limited industrial and agricultural activities. The movement of pollutants surrounding the plateau is often blocked by large mountains. The TP thus has a highly regional background atmosphere, which reflects sensitive feedback between changes in the local

environment and global climate change. Research on VOCs over the TP has been sporadic. For example, only a few studies have investigated the VOC levels at the Waliguan (Mu et al., 2007; Xue et al., 2013) and Menyuan (Zhao et al., 2020) background stations on the northeast of the TP, in Lhasa City (Yu et al., 2001; Guo et al., 2022; Yu et al., 2022a, 2022b; Ye et al., 2023), and at other sites (Li et al., 2017; Tang et al., 2022), and few studies have investigated specific organic pollutants,

such as persistent organic pollutants and polycyclic aromatic hydrocarbons (Liu et al., 2013; Chen et al., 2014; Wang et al., 2015, 2018; Sun et al., 2021) as well as carbonaceous aerosols (Yan et al., 2019, 2020).

The southeast of the TP, such as the Nyingchi area, has high vegetation cover; this region contains forest, river valley meadows, shrub land, and shrub steppes, and the land cover is considerably different from that in other regions of the TP. The southeast

of the TP is also a humid area with relatively high temperature and abundant precipitation because it is strongly influenced by the Indian monsoon. Therefore, the southeast of the TP is an ideal area for studying the natural VOCs and their potential influence given the transboundary and transcontinental transport of atmospheric pollutants from south Asia. Moreover, because of the progression of China's strategy to develop western areas of the country, Tibet has been undergoing rapid economic

development, including increases in its population, level of urbanization, and level of tourism activity; these changes have led to considerable energy demands, major changes in the region's traditional energy structure, a rapid rise in the number of vehicles, and increases in pollutant emissions. How and to what extent anthropogenic pollution emissions are affecting the local levels of VOCs are key research questions.

Within the framework of the "Field campaigns on the atmospheric chemistry over the Tibetan plateau: Measurement, processing, and the impacts on climate and air quality" (referred to as the @Tibet field campaigns), we carried out comprehensive measurement on budget of OH radicals, $O_3$, $NO_x$ and VOCs. From April to May 2021, online measurements of atmospheric VOCs were made at Lulang, Nyingchi (Linzhi) City with an alpine forested environment in the southeast of

the TP. In this paper, the resultant dataset was used to determine the major VOCs in the region, the main sources of these VOCs, and the contribution of anthropogenic emissions and natural life cycles to these VOCs.

## 2 Materials and Methods

### 2.1 Site description

The Lulang observation site (94.73°E, 29.76°N, 3326 m a.s.l.) is located in the yard of the Comprehensive Observation and Research Station of the Alpine Environment in Southeastern Tibet, Chinese Academy of Sciences. The station is located in a narrow mountain valley with a width of less than 1 km, and the surrounding mountains reach more than 1000 m higher than the valley (Figure S1). The bottom of the valley is covered in mountain meadows, spruce, and bare soil, whereas the mountains

are covered in bushes and dense spruce and pine trees, with almost full forest coverage. To the northwest of the observation site is a river confluence into the ditch on the east side of the national road. Zhaxigang Village (with a population of approximately 300 people) and Lulang Town (population of 1500) are 1.7 and 3.5 km to the south of the site, respectively. Nyingchi City (Bayi Town) is 39 km southwest, and Lhasa City is 350 km west of the site; both cities are separated from the

observation site by mountains. To the east of the site, National Road 318 runs from north to south, and there is a gas station 120 m to the southeast of the station. Traffic on the road is occasional, concentrated in the periods 8:00–10:00 and 12:00– 14:00. Because it was not peak tourist season during the observation period, relatively few tourist vehicles were on the roads,



and the sampling site was not strongly affected by high-intensity on-road emissions (Chen et al., 2014).

## 2.2 Measurements

Online VOC measurements were performed from April 4 to May 11, 2021, by using a low-temperature preconcentration sampling system (Pengyuchangya, Beijing, China) and a gas chromatography with flame ionization and mass spectrometry detectors (Shimadzu, Japan) analyzer with a time resolution of 1 h. Details on the apparatus can be found in Ye et al. (2023). In total, 101 VOCs were quantified: 28 alkanes, 14 alkenes, 16 aromatics, 28 halohydrocarbons, 13 OVOCs, acetylene, and acetonitrile. The $NO/NO_x$, $O_3$, and CO mixing ratios were determined using TE42itl, TE49i, and TE48itl commercial analyzers (ThermoFisher Scientific, MA, USA), respectively. The instruments were calibrated before and after a measurement was performed. The TE48itl CO analyzer was run with a 10-min zero check-in every 2 h. The temperature, humidity, wind speed, and wind direction were measured by a meteorological station at which two instruments were installed (the HMP155A, Vaisala, Finland, and 010C-1/020C-1, Metone, USA). The diurnal variations in the temperature, relative humidity, wind speed, and wind rose during the observation period are illustrated in Figure S2.

### 2.3 VOC reactivity calculation and source apportionment

The OH reactivity, ozone formation potential (OFP), and secondary organic aerosol potential (SOAP) of each VOC component were calculated to evaluate the VOC component's contribution to the formation of $O_3$ and SOAs. The overall OH reactivity of all measured VOCs was also summed and is referred to as VOC-$k_{OH}$ herein. The positive matrix factorization (PMF) model was used for VOC source apportionment. Details for these methods and their applications can be found in Ye et al. (2023).

### 2.4 OVOC source fitting based on photochemical age parameterization

One hypothesis of the PMF model is that the composition profile of pollutants does not change in the air—that is, that no chemical reaction occurs during the transport of pollutants—and that the concentration of pollutants does not change due to chemical reactions (Yuan et al., 2012). This hypothesis clearly entails an acute contradiction because some VOCs (e.g., OVOCs) are highly active and undergo rapid photochemical reactions. Therefore, the initial mixing ratio has been proposed to be a suitable substitute for the actual measured mixing ratio (Huang et al., 2020; Zheng et al., 2021). For a certain $VOC_i$ species, the initial mixing ratio $[VOC_i]_{t0}$ can be deduced as follows (McKeen et al., 1996; de Gouw et al., 2018):

$$[VOC_i]t = [VOC_i]_{t0} \times \exp{(-k_i[OH]\Delta t)} \tag{1}$$

where $[VOC_i]_t$ is the mixing ratio measured at time $t$, $k_i$ is the reaction rate constant of $VOC_i$ with OH radicals, $[OH]$ is the average concentration of OH radicals, and $\Delta t$ is the photochemical age of $VOC_i$. According to Equation (1), the hydroxyl exposure ($[OH]\Delta t$) can be determined if the initial mixing ratio is known. In this study, we used isoprene and its intermediate compounds, methacrolein (MACR) and methyl vinyl ketone (MVK), to determine $[OH]\Delta t$.

Isoprene + OH → 0.54 × (MVK+MACR), $k_1 = 1.00 \times 10^{-10}$ $cm^3$ molecule$^{-1}$ s$^{-1}$

MVK + OH → products, $k_2 = 2.0 \times 10^{-11}$ $cm^3$ molecule$^{-1}$ s$^{-1}$

MACR + OH → products, $k_3 = 2.9 \times 10^{-11}$ $cm^3$ molecule$^{-1}$ s$^{-1}$

$$\frac{[MVK+MACR]}{[Isoprene]} = \frac{0.32k_1}{k_2-k_1}\{1 - \exp[(k_1 - k_2)[OH]\Delta t]\} + \frac{0.23k_1}{k_3-k_1}\{1 - exp\ [(k_1 - k_3)[OH]\Delta t\}... \tag{2}$$

where $k_1$, $k_2$, and $k_3$ are the rate constants of the reactions of isoprene, MVK, and MACR with OH, respectively. On the basis of observed isoprene, MVK, and MACR mixing ratios (at time $t$), the hydroxyl exposure $[OH]\Delta t$ can be calculated using Equation (2).

A photochemical-age-based parameterization (PAP) method was used to quantify the contributions of anthropogenic emissions, anthropogenic generation of secondary pollutants, biological emissions, and background concentrations of OVOCs



(de Gouw et al., 2005; Liu et al., 2009; de Gouw et al., 2018; Huang et al., 2020). The PAP method is based on the following assumptions: (1) the emission of each OVOC is proportional to that of an inert tracer (such as CO, acetylene, or benzene); (2) OVOCs are chemically removed mainly through reactions with OH radicals; and (3) the scale of the biological source of OVOCs is directly proportional to the emissions of isoprene. On the basis of these three assumptions, the source composition of OVOCs can be expressed as follows:

$$[OVOC] = [OVOC]_{PA} + [OVOC]_{SA} + [OVOC]_{Bio} + [OVOC]_{BG}$$

$$= ER_{OVOC} \times [Benzene] \times \exp(-(k_{OVOC} - k_{benzene})[OH]\Delta t)$$

$$+ ER_{pre.} \times [Benzene] \times \frac{k_{pre}}{k_{OVOC} - k_{pre}} \times \frac{\exp(-k_{pre}[OH]\Delta t) - \exp(-k_{pre}[OH]\Delta t)}{\exp(-k_{benzene}[OH]\Delta t)}$$

$$+ ER_{bio} \times [Isoprene]_{t0} + [background] \tag{3}$$

With regard to the aforementioned equation, OVOC sources are considered to comprise primary anthropogenic emissions (denoted *PA*), anthropogenic secondary formation (*SA*), natural sources (*Bio*), and background (*BG*). $ER_{OVOC}$, $ER_{pre}$, and $ER_{bio}$ represent three emission ratios: that of OVOC emissions from the OVOC's primary source to benzene (the tracer) emissions, that of emissions of the anthropogenic precursor to the OVOC to benzene emissions, and that of OVOC emissions from the OVOC's biogenic source to isoprene (the natural tracer) emissions, respectively. The parameter $k_{pre}$ is the rate constant for the reaction of an anthropogenic precursor with OH radicals. A nonlinear regression analysis was performed using the concentrations of OVOCs, benzene, and isoprene; the rate constants for their reactions with OH radicals (denoted $k_{ovoc}$), and the hydroxyl radical exposure [OH]$\Delta t$. This analysis was conducted to estimate $ER_{OVOC}$, $ER_{pre}$, $ER_{bio}$, $k_{pre}$, and the background concentration (*background*).

## 3 Results and Discussion

### 3.1 Data overview and general characteristics of VOCs

The hourly mean total VOC (TVOC) mixing ratios were 3.87–33.1 ppbv (Figure S3); the average of these values was 11.72 ± 4.45 ppb, and the median value was 10.26 ppb. The average mixing ratios (±1σ) for alkanes, alkenes, aromatics, haloghydrocarbons, OVOCs, acetylene, and acetonitrile were 2.76 ± 2.52, 0.85 ± 0.62, 0.47 ± 0.38, 1.11 ± 1.07, 5.74 ± 4.93, 0.52 ± 0.44, and 0.27 ± 0.23 ppb, respectively, and these species accounted for 24%, 7%, 4%, 9%, 49%, 5%, and 2% of the total amount of VOCs, respectively (Figure 1a). The proportion for OVOCs was highest (49%), and the widest distribution range of mixing ratios was also obtained for these species; OVOCs contributed as much as 65% of VOC-$k_{OH}$ and 63% of the OFP because they have relatively high reactivity. Alkenes, which are also highly reactive, contributed 25% of VOC-$k_{OH}$ despite constituting only 7% of the TVOCs. Alkanes comprised 24% of the TVOCs but contributed only 4% and 5% of VOC-$k_{OH}$ and the OFP, respectively. Aromatics contributed only 6% of VOC-$k_{OH}$ and 14% of the OFP but made the largest contribution to the SOAP (89%).

The diurnal variation in OVOCs was considerably different from that in the other VOC species (Figure 1b and Figure S4); the level of OVOCs was high in the daytime (7.89 ± 3.02 ppb) but low at night (3.53 ± 1.80 ppb). A high OVOC level in daytime indicated that OVOC sources were present during the daytime; these sources must have been mainly photochemical conversion and biological emissions under strong solar radiation. The levels of alkanes, alkenes, and aromatics were found to clearly peak in the morning, similar to the levels of CO and $NO_x$; this finding indicated the impacts of anthropogenic emissions. The mixing ratios for alkanes and alkenes were higher in the daytime than at night, with daytime-to-nighttime ratios of 1.2 and 1.5, respectively. The main alkanes and alkenes were $C_2$–$C_5$ hydrocarbon compounds, which originate from vehicle-related and plant-related emissions. The mean mixing ratios of aromatics were similar in daytime (0.45 ± 0.31 ppb) and nighttime (0.51 ± 0.40 ppb).



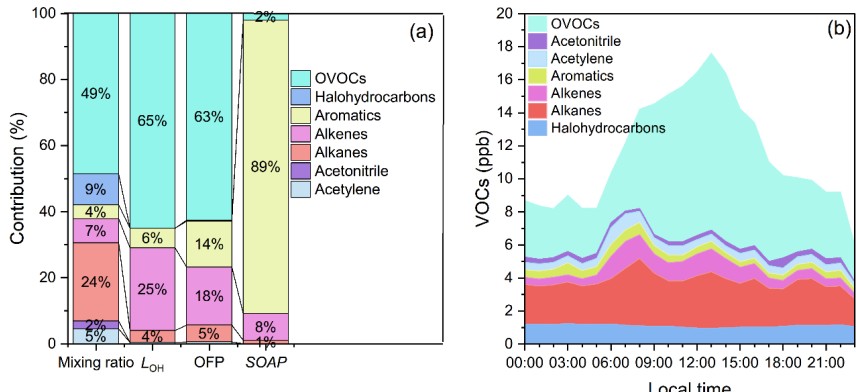

**Figure 1: (a) Contributions of various VOCs to the TVOCs, VOC-$k_{OH}$, the OFP, and the SOAP and (b) average diurnal variation in various VOCs.**

Figure 2 shows the 50 species with the highest mixing ratios and their corresponding VOC-$k_{OH}$, OFP, and SOAP. These 50 species accounted for 98.5% of the TVOCs. The mixing ratios of acetaldehyde and acetone were $1.96 \pm 1.71$ and $1.87 \pm 0.69$ ppb, and these species accounted for 16.7% and 16.0% of the TVOCs, respectively. Because of its high reactivity, acetaldehyde

made the largest contribution to VOC-$k_{OH}$ and the OFP. By contrast, the mixing ratios of ethane and acetylene were 1.46 and 0.52 ppb, respectively; ethane and acetylene ranked third and fourth and accounted for 12.4% and 4.5% of the TVOCs, respectively, but they made small contributions to VOC-$k_{OH}$ and the OFP due to their relatively low reactivity. Alpha-pinene ranked 25th (mixing ratio = $0.10 \pm 0.03$ ppb), but in terms of its contribution to VOC-$k_{OH}$ and the OFP, it ranked 5th (6.2%)

and 12th (2.8%), respectively. Additionally, the contribution of alpha-pinene to the SOAP was second only to that of aromatics. Although the mixing ratio of isoprene was only $0.02 \pm 0.01$ ppb and isoprene accounted for only 0.2% of the TVOCs, it made a contribution of 1.5% to VOC-$k_{OH}$ due to its very high $k_{OH}$. MACR and MVK, the secondary products of isoprene, had a total mixing ratio of 0.13 ppb and made the 10th-largest contribution (4.0%) to VOC-$k_{OH}$ and the 8th-largest contribution (4.3%) to

the OFP. Alkenes, acetaldehyde, acetone, and propanal also made contributions to the SOAP. Consequently, OVOCs and alkenes had important effects on the air at Lulang.

The mean TVOC mixing ratio was slightly lower than that determined in a previous study for the period May to June 2018 at Mt. Wudang in Hubei Province ($12.17 \pm 3.66$ ppb; Li et al., 2021). The average mixing ratio of 52 C4–C12 compounds was

176 2.84 ppb, which was lower than the value of $6.95 \pm 5.71$ ppb reported for Mt. Tai in June 2006 (Mao et al., 2009) and of 8.75 $\pm 5.76$ ppb reported for Gongga Mountain in Sichuan Province for the period January 2008 to December 2011 (Zhang et al., 2014b). For comparison, Table S1 presents the levels of major alkane, alkene, acetylene, and aromatic species reported in the literature with the levels determined in the present study. The mixing ratios of most species at Lulang (present study) were

180 close to those observed at the Menyuan atmospheric background station (Zhao et al., 2020) and Waliguan World Meteorological Organization Global Atmosphere Watch (GAW) station (Xue et al., 2013), which are both on the northeast of the TP. The mixing ratios of most species were approximately 10 times higher than those reported for the Antarctic and Arctic regions (Hellen et al., 2012; Pernov et al., 2021) but lower than those determined at the Lin'an regional GAW station in the

184 Yangtze River Delta, at rural stations of Tengyue in southwest China (Tang et al., 2009), and at Xianghe on the North China Plain (Yang et al., 2020).

High OVOC proportions have also been found at Menyuan (~50% in autumn 2013; Zhao et al., 2020), Nam Co (61%; Xu et al., 2023), and Lhasa City (52%; Ye et al., 2023) on the TP. The OVOC proportion discovered in the present study was higher





than that found in Beijing, where OVOCs were found to account for 32% of the TVOCs in the summer of 2016 (Gu et al., 2019). For comparison, Table S2 presents the major OVOC species levels reported in some studies. The acetaldehyde and acetone mixing ratios in the present study were 1.96 and 1.87 ppb, respectively, which were much lower than those reported for Beijing (Huang et al., 2020), Shanghai (Huang et al., 2008), and Guangzhou (Lyu et al., 2010) in China and for Monterrey

in Mexico (Menchaca-Torre et al., 2015). However, the present study's mixing ratio of acetaldehyde was higher than that determined in Shenzhen in 2018 (Huang et al., 2020), which has a similar latitude but a lower altitude than Lulang. As detailed in Table S1, although the mixing ratios of many hydrocarbons were lower in Lulang than in Xianghe (a rural site on the North China Plain) in the winter of 2017 (Yang et al., 2020), the mixing ratios of acetaldehyde and acetone were higher at Lulang

than at Xianghe (Table S2).

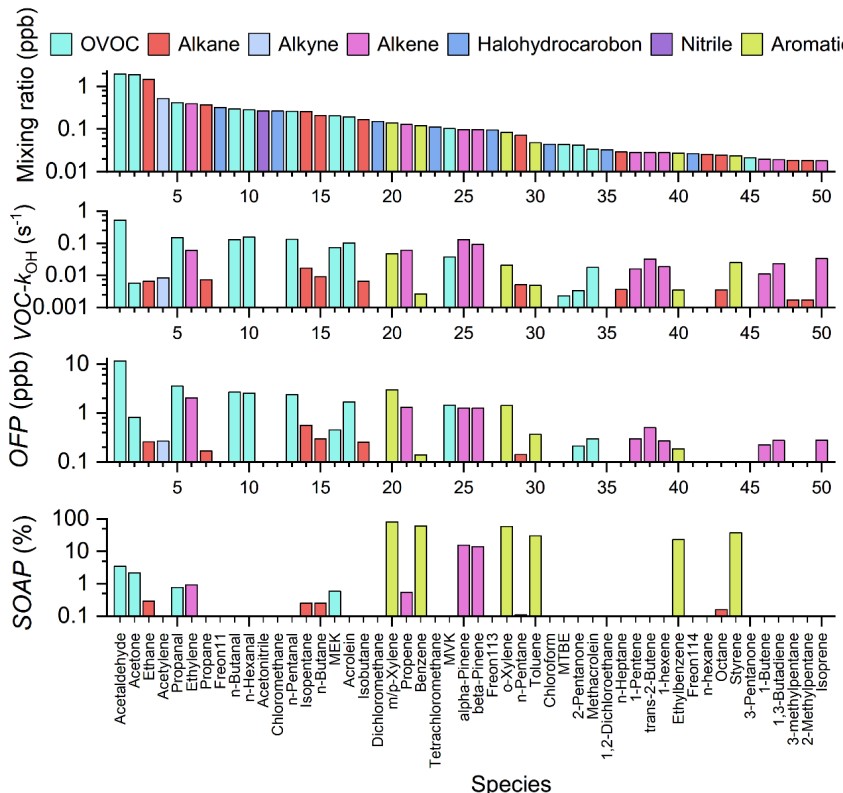

**Figure 2: Fifty species with the highest mixing ratio and their corresponding VOC-$k_{OH}$, OFP, and SOAP values. MEK: methyl ethyl**
**ketone; MVK: methyl vinyl ketone; MTBE: methyl tert-butyl ether.**

Figure 3 compares the levels of the main representative anthropogenic VOCs, biogenic VOCs, and OVOCs. The benzene level determined in the present study was slightly higher than that measured at some background stations but much lower than that
at other background stations and rural stations such as Gongga (Zhang et al., 2014), Lin'an (Tang et al., 2009), and Xianghe (Yang et al., 2020). The present study's mixing ratio of isoprene was also lower than that at most background and forest sites. However, the acetaldehyde and acetone levels were all higher than at most background sites and even comparable to those in some urban observations (Ho et al., 2002; Jiang et al., 2019). The daytime and nighttime (MVK+MACR)/isoprene ratios were
higher than those in most of the forest observations.



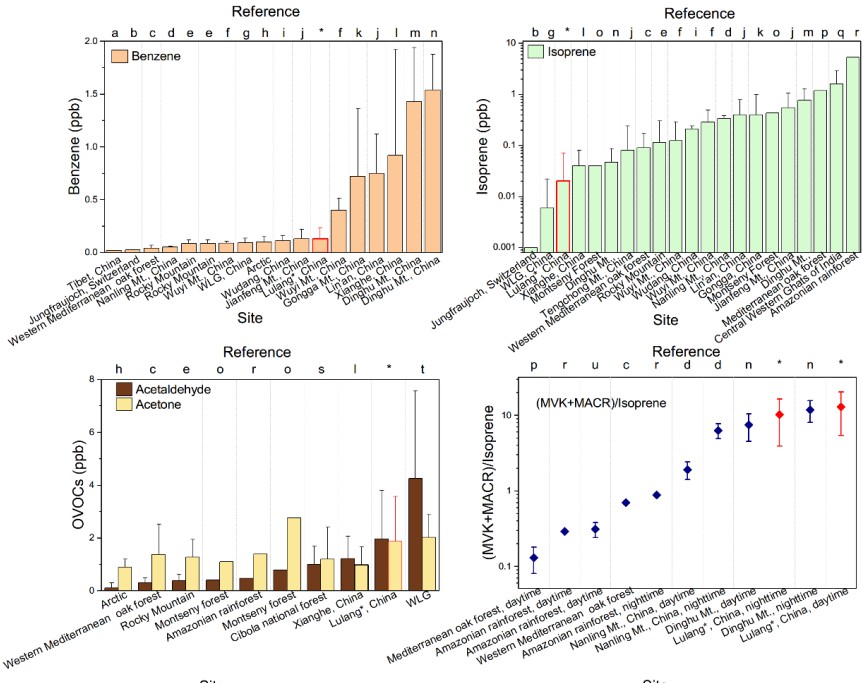

**Figure 3:** Comparison of the mixing ratios of benzene, isoprene, acetaldehyde, and acetone between Lulang and other sites (error
bars represent 1σ). a (Li et al., 2017); b (Legreid et al., 2008); c (Yáñez-Serrano et al., 2021); d (Gong et al., 2018); e (Benedict et al.,
2019); f (Hong et al., 2019); g (Xue et al., 2013); h (Hornbrook et al., 2016); i (Li et al., 2021a); j (Tang et al., 2009); k (Zhang et al.,
2014); l (Yang et al., 2020); m (Wu et al., 2016); n (Li et al., 2021b); o (Seco et al., 2011); p (Kalogridis et al., 2014); q (Tripathi et al.,
2021); r (Yáñez-Serrano et al., 2015); s (Villanueva-Fierro et al., 2004); t (Mu et al., 2007); u (Kuhn et al. 2007).

In summary, moderate VOC levels but a large contribution of OVOCs were discovered at Lulang and have also been found at
other sites on the TP. A relatively high OVOC proportion and high OVOC daytime levels on the TP are likely due to the
region's strong solar radiation and high atmospheric oxidative capacity (Chen et al., 2015; Lin et al., 2008), which facilitate
the secondary transformation of VOCs, thus leading to a high proportion of OVOCs. In addition, vehicle-related emissions
can be hypothesized to have some contribution.

## 3.2 Source apportionment and relative contributions

On the basis of their abundance, signal-to-noise ratio, and representativeness of certain VOC species in terms of source, 42
VOC species and 3 inorganic substances ($NO_x$, $NO_2$, and CO) were selected for PMF source apportionment. The optimal
number of decomposed factors was six. The source profiles and contributions of these six factors are illustrated in Figure 4,
and their diurnal variation is shown in Figure 5.

In Factor 1, large contributions were made by acetylene, benzene, acetonitrile, and chloromethane. Acetylene is a tracer of
incomplete combustion (Barletta et al., 2005; Zhang et al., 2019a), whereas acetonitrile and chloromethane are usually used as
tracers of biomass combustion (Chen et al., 2017; Koppmann et al., 2005). The mixing ratio of benzene was higher than that
of toluene (4:1), which is consistent with the characteristics of biomass combustion (Barletta et al., 2005; Liu et al., 2008).
Large contributions were also made by CO, ethane, propane, ethylene, and 1.3-butadiene, which are common substances in
combustion emissions. In this factor, the contributions of halogenated hydrocarbons, such as Freon11 and dichloromethane,
and acetone, which is more inert than aldehydes such as acetaldehyde, were also considerable. The aforementioned inert



compounds exist in the atmosphere as background; thus, Factor 1 represented VOCs originating from biomass combustion and regional background. The levels of the TVOCs in Factor 1 peaked at 6:00 and 18:00, which was related to the villagers' activities such as cooking, heating, and burning, and these levels were low at 13:00, which is the time when the boundary layer height is greatest.

In Factor 2, large contributions were made by aldehydes and ketones—such as acetaldehyde, acrolein, acetone, propanal, n-butanal, and n-pentanal—but not by methyl tert-butyl ether (MTBE). The diurnal variation plot indicated that the levels of these species were high in the daytime. This factor was related to photochemical production, which is a major source of carbonyl compounds (Mellouki et al., 2015; Zhang et al., 2019b). Large contributions were also made by isoprene, ethylene and propene, which can be strongly emitted by plants (Sindelarova et al., 2014) depending on photosynthesis, temperature, and solar radiation (Fuentes and Wang, 1999). The active isoprene, ethylene, and propene in air can be oxidized through a photochemical reaction to produce intermediate MACR, intermediate MVK, and other secondary OVOCs, which were also discovered to make large contributions in Factor 2. Because the TVOCs in Factor 2 had higher mixing ratios in the daytime than that at night, which was consistent with isoprene and OVOCs and was related to sunlight, Factor 2 could be classified as representing sunlight-impacted sources.

In Factor 3, the largest contribution was made by $NO_x$, and relatively large contributions were made by acetylene, ethane, acetaldehyde, and acetone. Much larger amounts of $NO_x$ are emitted by diesel vehicles than by gasoline vehicles (Li et al., 2020). Diesel vehicles carrying goods occasionally traveled on the nearby national road. The TVOCs in this factor peaked at 07:00 and 20:00 and thus had diurnal variation similar to that in $NO_x$. VOC mixing ratios were higher mainly when the wind was weak, reflecting the influence of local sources. Therefore, this factor was classified as representing diesel vehicle emissions.

In Factor 4, the main contributions were from C2–C5 alkanes, MTBE, and toluene, which are pollutants characteristic of motor vehicle exhaust (Liu et al., 2008; Cai et al., 2010; Chen et al., 2013; An et al., 2017; Mo et al., 2018; Song et al., 2021). MTBE, a typical tracer of gasoline vehicle emissions (Chang et al., 2006), made the largest contribution (almost 80%). The ratio of benzene to toluene (1:2) agreed with the emission characteristics of gasoline vehicles (Liu et al., 2008). In addition, isoprene, which is also emitted from motor vehicle combustion, made a contribution of 15% (Borbon et al., 2001). The diurnal variation in the TVOCs in Factor 4 was consistent with that in MTBE, isopentane, and related species. The Lulang site is close to National Road 318 and a gas station by the roadside and is thus influenced by gasoline vehicle exhaust and gasoline volatilization, resulting in variation in the daytime. Therefore, Factor 4 was classified as representing gasoline vehicle emissions.

In Factor 5, the largest contribution was that of monoterpenes at more than 80%, and an 18% contribution was made by isoprene. Monoterpenes mainly originate from the emissions of coniferous forests (Shao et al., 2001; Klinger et al., 2002; Li et al., 2019), whereas isoprene is mainly emitted from broad-leaved trees, which emit monoterpenes at lower levels. The diurnal variation in the TVOCs in Factor 5 was dominated by monoterpenes, for which the mixing ratios were higher at night than in the daytime; this finding was different from that for Factor 2. The observation site is surrounded by dense forest and is thus greatly influenced by natural emissions. Therefore, Factor 5 was classified as representing direct plant emissions.

In Factor 6, the main contributions were from ethylbenzene, xylene, and chloroform, which are commonly used as solvents (Yuan et al., 2010; Zheng et al., 2021). The contributions of CO and $NO_x$ were small. The TVOCs in Factor 6 had a lower mixing ratio in the daytime (0.54 ± 0.6 ppb) than at night (1.09 ±1.23 ppb). VOC mixing ratios were higher mainly when the wind was weak, reflecting the influence of local sources. Therefore, this factor was classified as representing solvent-related emissions.

Factor 1 (biomass combustion and background) and Factor 2 (sunlight-impacted) contributed 33% and 37% to the TVOCs, **16% and 53% to VOC-$k_{OH}$, 18% and 50% to the OFP, and 30% and 45% to the SOAP, respectively. Factor 5 (direct plant** emissions) made 10%, 16%, 10%, and 9% contributions to the TVOCs, VOC-$k_{OH}$, the OFP, and the SOAP, respectively. The contribution of transportation (Factors 3 & 4) was 15%, and that of solvents was 5% (Figure S5). Consequently, sun-



impacted VOCs played the most important role in the local atmospheric chemistry. Biomass burning was another key VOC source in the TP (Guo et al., 2022; Ye et al., 2023). The contribution of transportation may be a matter of concern.

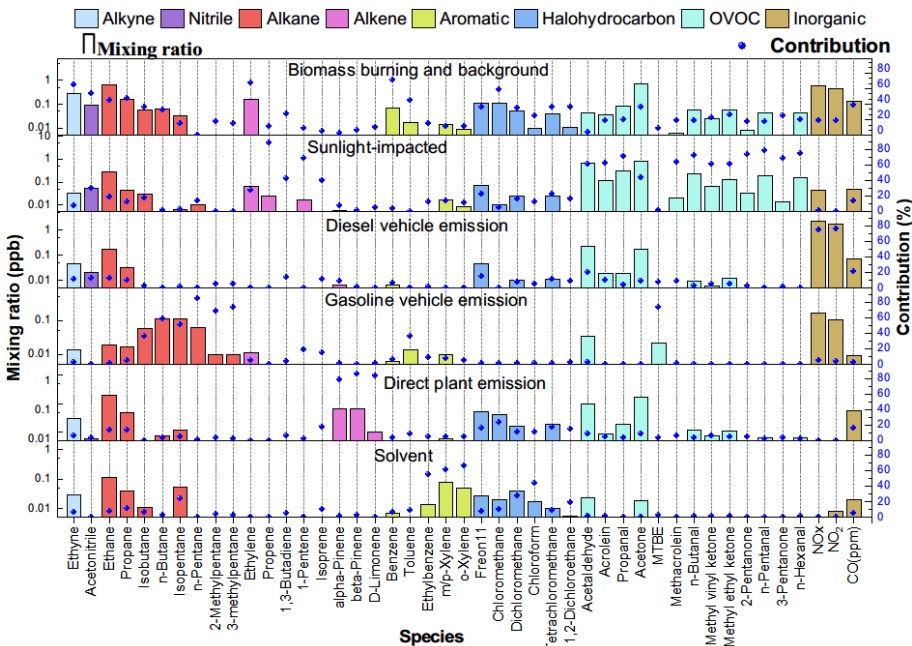

**Figure 4: VOC profiles and contributions in different factors after decomposition through PMF analysis**

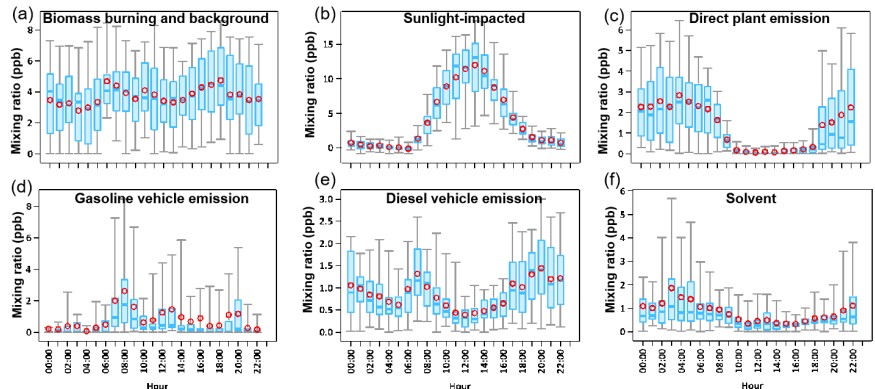

**Figure 5: Average diurnal variation in TVOCs in each factor. Bars represent the 25%–75% quantile, whiskers represent the 5% and 95% quantile, red circles represent the median, and blue horizontal lines represent the mean.**

### 3.3 OVOC source fitting through PAP

#### 3.3.1 OH exposure and initial isoprene mixing ratio

The [OH]$\Delta t$ and initial isoprene mixing ratios were calculated from the measured isoprene, MACR, and MVK mixing ratios; the results are plotted in Figure 6. The value of [OH]$\Delta t$ was in the range $8.9 \times 10^9$ to $5.3 \times 10^{10}$ molecules·cm$^{-3}$·s, with an average value of $(3.4 \pm 0.8) \times 10^{10}$ molecules·cm$^{-3}$·s. The calculated photochemical age was approximately 9.4 h and the



calculated lifetime of isoprene was 2.8 h when an average [OH] value of $1.0 \times 10^6$ molecules·cm$^{-3}$ was taken. The initial isoprene mixing ratios were much higher than the measured ratio, and the range of values for the ratio of initial-to-measured isoprene ratios was 5–150. Freshly emitted isoprene within the valley where our measurement site was located would not go through an aging process lasting for 9.4 h, given the rapid transport of fresh emissions within the valley (wind speed of 2 m/s vs. characteristic valley length of 40 km). The long photochemical age thus indicated the influence of the location. In the

daytime, a mountain-valley breeze brings aged air masses coming from places with a lower altitude relative to the measurement site.

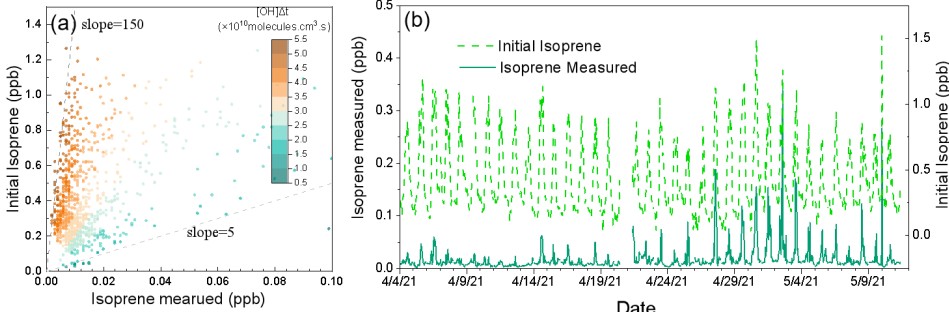

**Figure 6: (a) Scatter plot of the measured and initial isoprene mixing ratios, where the colors of dots represent hydroxyl exposure [OH]Δ$t$, and (b) time-series variation in the measured and initial isoprene mixing ratios.**

### 3.3.2 OVOC source fitting

After a nonlinear regression analysis was performed, the parameters $ER_{OVOC}$, $ER_{pre}$, $ER_{bio}$, and $k_{pre}$ and the background

concentration were fitted; the results are shown in Table S3. The OVOC sources and their relative contributions were estimated and are presented in Table 1. Biological sources contributed 62% of the amount of acetaldehyde, whereas the background contributed 30%. The contributions of background and biological sources to the amount of acetone were equal. According to the correlation coefficients, the fit was better for acetone than for acetaldehyde (Figure S6). Because the hydroxyl reaction rate

constant of acetaldehyde is approximately 100 times that of acetone, acetone is more inert and has a longer atmospheric lifetime, resulting in the background's contribution being higher. As detailed in Table 1, the fitting correlation coefficients ($r$) for the OVOCs other than acetaldehyde were all greater than 0.80, indicating a reasonable fit. Biogenic sources were found to make the dominant contribution to the amounts of most of the OVOCs, especially to the amounts of n-butanal, n-pentanal, and n-

hexanal. The contributions of biogenic sources to the amounts of OVOCs were much higher than those reported for Shenzhen in 2016 (Zhu et al., 2019), Beijing in 2018 (Huang et al., 2020), and Guangzhou in autumn and winter (Wu et al., 2020). This is expected because Lulang has a sparse population and dense forests and plant emissions make a large contribution to VOCs in this location.

**Table 1. OVOC source contributions (%), as determined through PAP fitting, and coefficient of correlation ($r$) between the fitted and measured OVOC mixing ratios.**

| Species | Primary Anthropogenic | Secondary Anthropogenic | Biogenic | Background | $r$ |
|---|---|---|---|---|---|
| Acetaldehyde | 7 | 1 | 62 | 30 | 0.45 |
| Acrolein | 0 | 5 | 80 | 15 | 0.85 |
| Propanal | 0 | 1 | 99 | 0 | 0.92 |
| Acetone | 0 | 10 | 45 | 45 | 0.85 |





| n-butanal | 0 | 0 | 100 | 0 | 0.92 |
|---|---|---|---|---|---|
| Methyl ethyl ketone | 0 | 5 | 79 | 16 | 0.92 |
| 2-Pentanone | 0 | 0 | 100 | 0 | 0.88 |
| 3-Pentanone | 0 | 0 | 100 | 0 | 0.91 |
| n-Pentanal | 0 | 0 | 100 | 0 | 0.91 |
| n-Hexanal | 0 | 0 | 100 | 0 | 0.80 |

### 3.3.3 Comparison of OVOC sources derived from PMF with PAP

At Lulang, acetaldehyde and acetone were discovered to be the two most abundant OVOCs, and the PMF-based source apportionment indicated that sunlight-impacted sources made contributions of 61% and 44% to the amounts of these OVOCs, respectively. These OVOCs thus originated mainly from plant emissions and secondary photochemical formation under solar radiation. This result was consistent with that obtained through PAP, which estimated 62% and 45% contributions from plant

sources to acetaldehyde and acetone, respectively (Table 1). Figure 7 illustrates the diurnal variation in the contributions of OVOC sources for the results derived from (a) PMF apportionment and (b) PAP fitting. The two sets of results were very similar, especially in their conclusion that biological sources (contribution: 65% vs. 67%) were the largest sources of VOCs in Lulang (Figure S6). Biomass burning sources were closely related to the background because biomass burning is common on

the whole TP, as also reported by Guo et al. (2022) and Ye et al. (2023). Using the PMF method resulted in a greater contribution from primary anthropogenic emissions but a smaller contribution from the background than using the PAP approach did (Figure S7). Generally, the PAP method underestimated the levels of OVOCs at approximately 12:00 and overestimated those in the morning and evening.

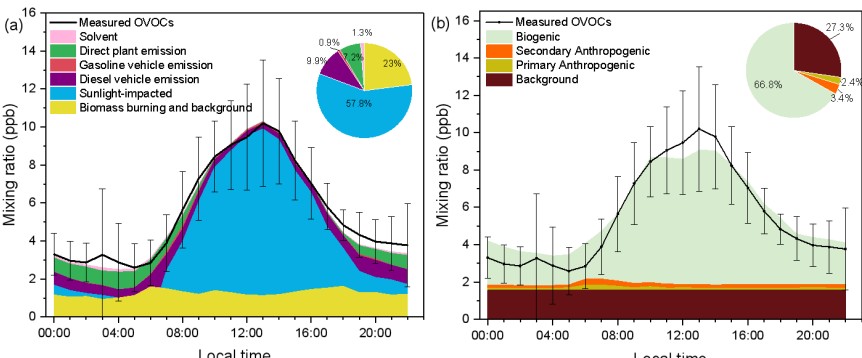

**Figure 7: Diurnal variation in OVOC sources derived from (a) PMF apportionment and (b) PAP fitting.**

### 4 Conclusions

Online measurement of atmospheric VOCs was conducted from April 4 to May 11, 2021, in Lulang, a site on the southeastern part of the TP that has high vegetation cover and strong ultraviolet radiation, to understand the sources and behaviors of VOCs in that location. In total, 101 VOC species were recognized and quantified: 28 alkanes, 14 alkenes, 16 aromatics, 28 halocarbons, 13 OVOCs, acetylene, and acetonitrile. The mixing ratio of the TVOCs was as low as $11.72 \pm 4.45$

340     ppb, indicating a background level in Lulang. OVOCs accounted for the largest proportion (49%) of the TVOCs, and the levels of OVOCs exhibited typical diurnal variation with high values in the daytime and a peak at approximately 12:00, a feature that has also been found in other areas on the TP. Due to their high reactivity, OVOCs contributed 65% and 63% to VOC-$k_{OH}$ and the OFP, which indicated that OVOCs have a strong influence on atmospheric chemical processes on the TP. Among the



OVOCs, the two most abundant species were acetaldehyde and acetone, for which the mixing ratio was 1.96 ± 1.71 and 1.87 ± 0.69 ppb, respectively.

The PMF model revealed six groups of sources of VOCs: sunlight-impacted sources, biomass combustion and background, direct plant emissions, diesel vehicle emissions, gasoline vehicle emissions, and solvent evaporation sources, with relative

contributions to the TVOCs of 37%, 33%, 10%, 9%, 6%, and 5%, respectively. The mixing ratio was highest for sunlight-impacted sources, and these sources also made the largest contributions to VOC-$k_{OH}$ (53%), the OFP (50%), and the SOAP (45%). The diurnal variation in the TVOCs with sunlight-impacted sources had a similar pattern to that for OVOCs and a different pattern to the TVOCs with other direct plant emission sources; these TVOCs were mainly monoterpenes and had low

mixing ratios in the daytime but high values at night. Sunlight-impacted and direct plant emission sources, both related to plant sources, together made a contribution of 47% of the TVOCs and 65% of the OVOCs. The OVOC sources determined through fitting with the PAP method also indicated that biogenic sources made the largest contribution to OVOCs (67%) and that OVOCs from these sources clearly peaked in the midday. The results obtained using the two methods were thus similar, and

the conclusions drawn regarding the importance of biological sources to VOCs in Lulang were consistent. Biomass burning sources were closely related to background VOCs because biomass burning is common across the whole TP, and these sources made the second greatest contribution to the TVOCs. Vehicle emissions are potentially an important contribution to the TVOCs and are a matter of concern. Our results have improved our understanding of the VOCs present in the region, and the findings

may be significant for the field of tropospheric chemistry.

**Author contribution**

C.Y. and W.L. designed the research. S.G., J.W. C.Z., and Y.C. carried out the field measurements. S.G. performed

data analysis and interpreted the data. C.Y., S.G., W.L., and J.W. prepared the manuscript with contributions from all co-authors.

**Competing interests**

The authors declare that they have no conflict of interest.

**Acknowledgement**

This work was supported by the National Natural Science Foundation of China (Grants No. 42375088, 21876214) and the Second Tibetan Plateau Scientific Expedition and Research Program (Grants No. 2019QZKK0604).

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
