# Peer review of "Surprisingly high levels and activity contributions of oxygenated volatile organic compounds on the southeast of the Tibetan Plateau"

_EGUsphere, 2024_

## Author Comment (AC1)

Reply on RC1

Thank you very much for your valuable comments and suggestions on our manuscript. We appreciate the time and effort you have invested in reviewing our work.

The manuscript, "Surprisingly high levels and activity contributions of oxygenated volatile organic compounds on the southeast of the Tibetan Plateau" based on the VOCs online observation experiment at the South-East Tibetan plateau Station for integrated observation and research of alpine environment in Lulang. It is 101 VOCs species were detected, including 13 OVOCs. The authors analyzed the contribution of OVOCs to VOCs concentration, OH• reactivity and OFP in the southeastern Tibetan Plateau. At the same time, PMF and PAP source analysis methods were combined to explain the maximum possible OVOCs contribution sources. However, there are doubts about the method used by the author to analyze OVOCs species by GS-MS/FID. The concentration of OVOCs obtained by GS-MS/FID method currently has a large uncertainty, so it is impossible to determine whether the conclusion obtained in this paper is correct. Therefore, the publication of this paper cannot be accepted, with deep regret.

Response:Regarding the concern about the method used to analyze OVOCs species by GC-MS/FID, we would like to clarify and provide additional information.

Firstly, the online GC-MS/FID analyzer we used is a mature and commercially available instrument which has been developed and utilized in field campaigns over the years (Wang et al., 2014). Moreover, the use of GC-MS for reporting measured OVOCs is not novel, as similar instruments have been widely utilized in various observations (Geir Legreid et al., 2007a, 2007b; X.P. Lyu et al., 2016; Wu et al., 2020; Yunfeng Li et al., 2021). To ensure the instrument's stability and the accuracy of our data, we implemented stringent quality control and assurance measures throughout the analysis process.

Secondly, regarding the VOCs and OVOCs data collected at the Lulang Observation Station, we have conducted data comparisons with other instruments, such as PTR-MS, used by other research teams for synchronized observations. The consistency of the data is satisfactory, e.g. a correlation coefficient (R) of 0.82 for acetone, further reinforcing the reliability of our measurements.

We hope that these clarifications and additional information address your concerns. Once again, we appreciate your valuable input and look forward to any further suggestions that could enhance our work.

Reference:

Wang, M., Zeng, L., Lu, S., Shao, M., Liu, X., Yu, X., Chen, W., Yuan, B., Zhang, Q., Hu, M., and Zhang, Z.: Development and validation of a cryogen-free automatic gas chromatograph system (GC-MS/FID) for online measurements of volatile organic compounds, Anal. Methods, 6, 9424–9434, https://doi.org/10.1039/c4ay01855a, 2014.

Geir Legreid, Stefan Reimann, Martin Steinbacher, Johannes Staehelin, Dickon Young, and Konrad Stemmler. Measurements of OVOCs and NMHCs in a Swiss Highway Tunnel for Estimation of Road Transport Emissions, Environ. Sci. Technol., 41, 20, 7060–7066, 2007,https://doi.org/10.1021/es062309

Geir Legreid, Jacob Balzani Lo¨o¨v, Johannes Staehelin, Christoph Hueglin, Matthias Hill, Brigitte Buchmann, Andre S.H. Prevot, Stefan Reimann, Oxygenated volatile organic compounds (OVOCs) at an urban background

site in Zu¨rich (Europe): Seasonal variation and source allocation, Atmospheric Environment, 41: 8409–8423, 2007b

X.P. Lyu, N. Chen, H. Guo, W.H. Zhang, N. Wang, Y. Wang, M. Liu,Ambient volatile organic compounds and their effect on ozone production in Wuhan, central China, Science of The Total Environment, 541, 200-209, 2016.

Wu, C., Wang, C., Wang, S., Wang, W., Yuan, B., Qi, J., Wang, B., Wang, H., Wang, C., Song, W., Wang, X., Hu, W., Lou, S., Ye, C., Peng, Y., Wang, Z., Huangfu, Y., Xie, Y., Zhu, M., Zheng, J., Wang, X., Jiang, B., Zhang, Z., and Shao, M.: Measurement report: Important contributions of oxygenated compounds to emissions and chemistry of volatile organic compounds in urban air, Atmos. Chem. Phys., 20, 14769–14785, https://doi.org/10.5194/acp-20-14769-2020, 2020.

Yunfeng Li, Rui Gao, Likun Xue, Zhenhai Wu, Xue Yang, Jian Gao, Lihong Ren, Hong Li, Yanqin Ren, Gang Li, Chuanxian Li, Zeliang Yan, Ming Hu, Qingzhu Zhang, Yisheng Xu, Ambient volatile organic compounds at Wudang Mountain in Central China: Characteristics, sources and implications to ozone formation, Atmospheric Research, 250, 105359, 2021, https://doi.org/10.1016/j.atmosres.2020.105359.

Specific comments

EPA TO15 and TO11A standard methods specify the analysis methods of NMHCs and carbonyl compounds in atmosphere respectively. When GS-MS/FID is used to analyze highly polar carbonyl compounds, there is a large uncertainty. The same adsorption column is used to detect OVOCs and other VOCs species, and the adsorption effect of the column adsorbed other VOCs is not good for polar OVOCs. Meanwhile, humidity and wall effect will make the GC-MS/FID method very susceptible to the detection of OVOCs species. Secondly, GS-MS/FID cannot detect formaldehyde, which accounts for a large proportion of OVOCs in daily detection. And the absence of formaldehyde may lead to the underestimation of the contribution of OVOCs to VOCs concentration and OFP, which has great limitations on the analysis of the nature of OVOCs species. Therefore, the current GC-MS/FID method for detecting OVOCs species is not reliable. The data thus obtained cannot lead to correct conclusions.

Response: Thank you for your further comments and suggestions. We appreciate your feedback and have addressed the points raised below.

Regarding the concerns about the reliability of the GC-MS/FID method for analyzing OVOCs species, we would like to clarify our instrument parameters and method settings. Our GCMS instrument parameters and method settings for this study are based on our previously published article in Atmos. Chem. Phys. (23, 10383–10397, 2023; https://doi.org/10.5194/acp-23-10383-2023), where we used the same instrument and method.

In our GCMS/FID setup, we employed two different chromatographic columns for separation. The PLOT Al2O3 chromatographic column (15 m × 0.32 mm inner diameter (i.d.) × 3 μm; J&W Scientific, USA) was used for the separation of 13 C2-C5 hydrocarbons, while the moderately polar DB-624 column (60 m × 0.25 mm i.d. × 1.4 μm; J&W Scientific, USA) was used for the separation of other VOCs, including OVOCs. Before the VOCs entered the chromatographic columns for separation, water, $CO_2$, and $O_3$ were removed to minimize interference in the analysis.

During the observation period, we performed multi-point calibrations using standard samples, and the linear correlation ($R^2$) for OVOC species was greater than 0.99 (Figure 1). Additionally, a daily span check was conducted (Figure 2). Although humidity and wall effects can indeed introduce some interference in measurements, our rigorous quality control measures have helped to reduce the uncertainty in OVOC measurements to a certain extent.

Regarding formaldehyde, GCMS method was not quantitative for its analysis. The absence of formaldehyde does indeed lead to an underestimation of the contribution of OVOCs. Typically, the concentration of formaldehyde in the atmospheric environment is slightly higher than that of acetaldehyde. Based on the VOCs data obtained during our measurements at Lulang, we found that the contribution of OVOCs, excluding formaldehyde, to concentration and reactivity is already significant. This does not affect our current conclusions and motivates us to further investigate atmospheric OVOCs in the future.

In the revised manuscript, we will incorporate additional measurement details and engage in a thorough discussion regarding the absence of other oxygenated volatile organic compounds (OVOCs), particularly formaldehyde.

[Figure]

Figure 1. Multi-point calibration results of OVOCs

[Figure]

Figure 2 Time-series variations in daily span check results

In lines 72-76, when describing the situation around the observation site, the author mainly introduced the sites tens of kilometers away. This description does not seem to highlight the point, nor does it play a significant role in understanding the origin of OVOCs species at the Lulang It is suggested that the author consider the description here and make changes.

Response:    The observation site is located in the forested regions over an area of 2.46 million hectares, accounting for 53% of the total land area and with a forest coverage rate is 46%. Other than the limited pollutants' emissions of the 318 national highway running through it, the most nearby anthropogenic emission source might be the nearby towns and cities. Our original intention in introducing these distant populated areas was to emphasize the remoteness and sparsity of the Qinghai-Tibet Plateau, as well as to suggest that anthropogenic emissions from these regions, although distant, may contribute to the background VOCs concentrations. Additionally, we included this description to provide a broader understanding of this lesser-known area, aiming to familiarize readers with its unique characteristics. We will rewrite and add some information in the revised manuscript.

Lines 160-173 describe the dominant species in no obvious order or logic, but the conclusion of the paragraph in line 173 states that OVOCs and alkenes have an important effect on the air of Lulang. It is suggested that OVOCs and other VOCs species should be explained in two parts based on the conclusion.

Response: We have rewritten the paragraph to ensure a logical flow of information and better align with the conclusion stated in Line 173. Here is a revised version:

"Figure 2 presents the 50 species with the highest mixing ratios, along with their corresponding VOC-$k_{OH}$, OFP, and SOAP values. These 50 species collectively accounted for 98.5% of the TVOCs. Among them, acetaldehyde and acetone emerged as significant contributors, with mixing ratios of $1.96 \pm 1.71$ and $1.87 \pm 0.69$ ppb, respectively. These two species alone accounted for 16.7% and 16.0% of the TVOCs. Acetaldehyde, due to its high reactivity, made the largest contribution to VOC-$k_{OH}$ and the OFP.

In contrast, ethane and acetylene, despite having mixing ratios of 1.46 and 0.52 ppb, respectively, and ranking third and fourth among the 50 species, contributed relatively little to VOC-$k_{OH}$ and the OFP due to their lower reactivity. Other species, such as alpha-pinene, also stood out. Although its mixing ratio was $0.10 \pm 0.03$ ppb, ranking 25th, it made significant contributions to VOC-$k_{OH}$ (5th, 6.2%) and the OFP (12th, 2.8%).

Additionally, alpha-pinene's contribution to the SOAP was second only to aromatics. Isoprene, despite its low mixing ratio of $0.02 \pm 0.01$ ppb and accounting for only 0.2% of the TVOCs, made a notable contribution of 1.5% to VOC-$k_{OH}$ due to its high reactivity with $k_{OH}$. MACR and MVK, secondary products of isoprene, with a combined mixing ratio of 0.13 ppb, were also significant, contributing 4.0% to VOC-$k_{OH}$ and 4.3% to the OFP.

Overall, the analysis reveals that OVOCs and alkenes, including acetaldehyde, acetone, alpha-pinene, and isoprene, play a crucial role in shaping the air quality at Lulang. Their combined effects on VOC-kOH, OFP, and SOAP underscore their importance in the region's atmospheric chemistry."

In lines 174-185, in the comparison between this experiment and other background sites or ordinary sites, the comparison logic is rather confused, and I can't see what the author wants to highlight at last. It is suggested that the author clarify the description logic, rearrange the content, and summarize the corresponding conclusion.

Response: To clarify the description and rearrange the content, we have revised the paragraph as follows:

"The mean TVOC mixing ratio observed in this study was compared with previous observations in mountain sites Table S1 presents a comparison of the levels of major alkane, alkene, acetylene, and aromatic species reported in the literature with those determined in the present study. Our comparison results suggest high altitude and low temperature might account for relatively low TVOC observed in this study. Specifically, our observation was slightly lower than that previous measurement conducted at Mt. Wudang in Hubei Province during May to June 2018 ($12.17 \pm 3.66$ ppb; Li et al., 2021). Furthermore, the average mixing ratio of 52 C4–C12 compounds observed at Lulang was lower than the values reported for Mt. Tai in June 2006 ($6.95 \pm 5.71$ ppb; Mao et al., 2009) and Gongga Mountain in Sichuan Province for the period January 2008 to December 2011 ($8.75 \pm 5.76$ ppb; Zhang et al., 2014b). However, it is evident that the levels observed in this study were close to those reported at the Menyuan atmospheric background station (Zhao et al., 2020) and the Waliguan World Meteorological Organization Global Atmosphere Watch (GAW) station (Xue et al., 2013), both located on the northeast of the Tibetan Plateau. The overall TVOC mixing ratio in these observations roughly follows the rule of "high temperature and more abundant VOCs". As our colleagues extended PTR-MS measurement in May, BVOC mixing ratio almost doubled as compared with PRT-MS observation in April during our campaign period.

TVOC mixing ratios in these maintain sites were approximately 10 times higher than those reported for the Antarctic and Arctic regions (Hellen et al., 2012; Pernov et al., 2021), while substantially lower than those determined at the regional background site, such as Lin'an regional GAW station in the Yangtze River Delta, rural stations in Tengyue in southwest China (Tang et al., 2009), and Xianghe on the North China Plain (Yang et al., 2020).

In summary, our observation site features a typical atmosphere of subalpine coniferous forest climate with slightly anthropogenic emission perturbation"

In lines 303-315, after completing the fitting of photochemical aging parameters, the author only analyzed acetaldehyde and acetone, and did not mention other species. It is suggested that the author supplement the reasons why these two species were discussed separately, or supplement the analysis of other species.

Response: The reason we focused our analysis on acetaldehyde and acetone is that they are among the most abundant and widely distributed species in the atmosphere, making them highly representative. Their high abundance and diversity of sources, including both anthropogenic and natural origins, make them key compounds in atmospheric chemistry.

While other OVOCs may have lower relative abundances, their sources, as our modeling suggests, are more limited and primarily natural. Given the scope and focus of our study, we chose to prioritize the analysis of acetaldehyde and acetone, which are not only more abundant but also play a crucial role in atmospheric photochemical processes.

We agree that a more comprehensive analysis of other OVOCs would provide additional insights. In the revised manuscript, we will emphasize the rationale behind selecting these two species, while highlighting other photochemical tracers, such as MVK.

Technical comments

In Line 16 of the article, ";" in (TVOCs; average level of $11.7 \pm 4.4$ ppb) looks some strange. It is suggested that the author to change.

Response: The revised sentence will read: "The 13 OVOCs detected accounted for 49% of the total VOCs (TVOCs), with an average level of $11.7 \pm 4.4$ ppb."

There are some formatting problems in the illustrations of the article, such as the English font in the figures does not seem to use the New Roman font, and the scale lines of the picture frame sometimes face inward and sometimes face out, so it is suggested that the author should unify to make the graphics more beautiful. Secondly, the vertical axis label fonts of sub-figure 1 in Figure 4 overlap, so it is suggested that the author modify them.

Response: accepted.

In the reference part of the article, the subscript format of "O3" should also be set. It is suggested that the author check and refer to all literatures and unify the format.

Response: accepted.

---

## Author Comment (AC2)

We thank for the constructive comments and suggestions. We revised our manuscript according to the comments and suggestions. The following list the point-to-point response to the comments.

The authors of the manuscript, "Surprisingly high levels and activity contribution of oxygenated volatile organic compounds on the southeast of the Tibetan Plateau", present observed characteristics of VOCs composition in a Tibet field campaign at Lulang, and investigate the sources of different VOC species using PMF and PAP approach. It is an interesting project, the outcome of which will definitely strengthen the understanding of atmospheric composition, especially in terms of OVOCs, in Tibetan Plateau. I appreciate the efforts the authors have made in conducting the in-situ measurements, analyzing the observational data, and performed the source apportionment. However, there are key questions need to be addressed before the manuscript become a qualified scientific article of Atmospheric Chemistry and Physics.

General comments:

1. The current Introduction provides very limited information on the innovation of the authors' study. While VOCs are important air pollutants affecting atmospheric chemistry and human health, the authors are expected to provide more specific information on what makes their study a unique one, instead of spending most words in general introductions which are commonly knew. As the authors listed, there have been a few studies investigating OVOCs at different sites over the TP. What are the differences between their study and the previous ones? What innovative findings can they provide to advance the understanding of OVOCs at TP? Unfortunately, the audience cannot tell according to the authors' descriptions. The VOC observations are so hard-won, and hopefully they can be well presented.

Response: we revise the Introduction section to provide more specific information on the novelty of our approach, the differences between our study and previous ones, and the innovative findings we aim to contribute.

2. As the authors have mentioned, the PMF model includes a hypothesis that the composition profile of the studied air pollutants does not change in the air. As OVOCs are highly active, the PMF model may not be a quit suitable approach for source attribution of OVOCs. The authors suggest that they have introduced the initial mixing ratio to address this problem for a certain VOC species. Have this correction been applied to the concentrations of all the species included in the source apportionment? To what extent this mismatching problem can be fixed? The description is not clear so the related results presented by the authors are questionable from the perspective of audience. Even though the authors have obtained similar source attribution results from PMF and PAP methods, I still suggest them to consider whether the PMF part of discussion should be included in the manuscript if they cannot address the problem correctly.

Response: We agree that the PMF model has inherent limitations in accounting for the chemical losses of VOCs during transport, which can introduce errors into the source apportionment results.

To address this concern, scientists have introduced the concept of the initial mixing ratio, using either the photochemical age method or the isoprene oxidation method, which can help mitigate this weakness to some extent. However, the initial mixing ratio is not directly measured but is calculated. These calculations rely on several assumptions, including that all major VOC species are emitted from the same source, that other emission sources are negligible during transport, and that chemical losses of VOCs are primarily due to reactions with OH radicals, while reactions with $NO_3$ radicals and other factors are ignored. These assumptions may not fully align with real-world conditions, introducing uncertainties into the calculated initial mixing ratios.

In our study, we did not apply the initial mixing ratio directly in the PMF model. Instead, we incorporated this approach into the PAP methods specifically designed for OVOC source apportionment. The similar results obtained from both the PMF and PAP methods suggest that, despite their differences, these approaches provide comparable insights into the sources of VOCs in the studied region. Importantly, the consistent findings from both methods indicate that the major VOC sources in the area are relatively stable.

We will add the discussion on above questions in the revised manuscript.

3. As for the PAP method, the authors use another assumption that the OVOC emission is proportional to that of an inert tracer. There are observational evidence suggesting the correlation between the emissions of secondary organic VOCs with these inert traces (e.g., CO and benzene) from traditional sector (e.g., transportation) and volatile chemical products (e.g., pesticides, coatings, inks, cleaning agents). Do these emissions are important sources of VOCs over the region of the authors' interest? Is there any observational evidence proving the rationality of this hypothesis in estimating OVOC emissions from other sectors? All the related queries should be addressed, otherwise the source attribution results cannot be logically convinced unfortunately.

Response: We have just applied the PAP method as it has been previously developed and utilized by many other studies. We agree that there is a need for consideration of the emission sources and their relationship to inert tracers in our region of interest. Currently, we do not have real ratios between anthropogenic VOC emissions and CO in our study area. As such, we have relied on the established approach within the PAP method. In the PAP method, emissions are represented using broad categories such as primary anthropogenic emissions, anthropogenic secondary formation, natural sources, and background, without specifying specific sources. We agree that the uncertainties associated with this approach should be further discussed in our analysis, and we will include uncertainty analysis in our revised manuscript. In addition, we will add related background introduction and judgement to illustrate that the results of source apportionment are consistent with our cognition of the local environment in the revised manuscript.

4. Given the shortcomings of the approaches that the authors have used in tracking the sources of VOCs, discussions on the uncertainty of their results should be necessary to show the audience the significance and representative of this study.
Response: We will intensify the discussion of uncertainty.

Specific comments:

L33: It would be better to use 'both' than 'whether' in this sentence.
Response: accepted.

L34: 'Activity' is a countable noun, so it would be good to use plural here.
Response: accepted.

L54: Transport from south Asia is not 'transcontinental' transport.
Response: corrected.

L68-79: There are geographical information on so many different sites there. It would be good to show a map figure so that the audience can tell the relative location of these names more directly.
Response: we will add information directly in Figure S1.

L77: Please add "LST" after the two time periods
Response: accepted.

L84: It would be good to include the name of each VOC species in Supporting Information so that the audience can get what VOC species are included in each group in this study.
Response: accepted. We added a list in the Supplementary materials.

L95: It would be good to introduce the approach briefly in the manuscript even though it has been used in other related works before.
Response: The PMF method is extensively employed in VOCs research and its description can be easily found elsewhere; therefore, in the interest of brevity, I believe it can be omitted from this text. Besides, the referenced literature is openly accessible.

L106-107: Please cite the related references here.
Response: Accepted.

L128: Please explain how the emissions of OVOC and benzene can be obtained.
See response 3.

L152: It seems that the Lulang site is not an urban site according to the descriptions in Introduction section, but the observed CO and NOx still show distinct urban diurnal variations. The results are kind of confusing.
Response: Despite not being a city station, the Lulang site is still subject to human activity, such as biomass burning and traffic emission for specific time periods.

L156: It is hard to tell the diurnal variations of each VOC species in Figure1b, since it shows the accumulated concentrations. The authors may consider to use a more direct way to

present this.

Response: In the Supplementary materials, Fig. S2 is another way to present the diurnal variations of different VOC species.

L174-196: I appreciate the efforts that the authors have made to compare their observations with literature studies. The results should be interesting. However, more innovative analysis should be conducted to make the manuscript more than an observational report. Unfortunately, the descriptions here are just another way to present Table S1. Questions (e.g., what cause the observed differences between this study and previous ones? Why the authors' measurements matter a lot in VOC observational studies?) that the audience may be interested in are not mentioned at all.

Response: Thanks for your suggestion. We will revise these sentences in the revised paper.

L211: The authors suggest there is a surprisingly high levels of OVOCs observed at Lulang, however, the observed concentrations seem not that 'surprising' compared to literature studies in Figure 3. The authors would make more efforts in presenting why they think their findings are surprising. Otherwise, they may want to revise the misleading tile of the manuscript.

Response: Upon further consideration, we agree that the concentrations themselves may not be exceptional when compared to those reported in other literature studies, as shown in Figure 3. However, our intention was to emphasize the unexpected significance of OVOCs as a contributor to the overall VOC levels and their associated activities on the southeast of the Tibetan Plateau. In light of this, we have revised the title to remove the word "Surprisingly," and it now reads: "High Levels and Activity Contributions of Oxygenated Volatile Organic Compounds on the Southeast of the Tibetan Plateau."

L271: Are there many emissions from solvent sources in Lulang?
In Tibet, people usually live along rivers due to the mountainous terrain. There is a small village near the observation point, so there will be solvent use or volatilization. However, as a result of source analysis, this is only a small fraction of the VOC.

Response: In Tibet, people usually live along rivers due to the mountainous terrain. There is a small village near the observation point, so there will be solvent usage or volatilization. However, as a result of source analysis, this is only a small fraction of the VOC.

L296: Please explain how the mountain-valley can affect the VOC concentrations at Lulang.

Response: When sunrise, the wind speed is up (Figure S2C), it transports the aged air to the station.

---

## Author Comment (AC3)

We thank for the constructive comments and suggestions. We revised our manuscript according to the comments and suggestions. The following list the point-to-point response to the comments.

Comment #3

The manuscript titled "Surprisingly high levels and activity contributions of oxygenated volatile organic compounds on the southeast of the Tibetan Plateau" by Guo et al. reports findings from the @Tibet field campaigns 2021. The study focuses on oxygenated volatile organic compounds (OVOCs), which are significant for tropospheric chemistry due to their roles as precursors to free radicals. The research found high levels of OVOCs in Lulang, a region on the southeast of the Tibetan Plateau characterized by high vegetation cover and intense solar ultraviolet radiation. The study detected 13 OVOCs accounting for 49% of the total VOCs, with notable diurnal variation peaking at noon. These compounds contributed significantly to VOC reactivity and ozone formation potential. Source apportionment using positive matrix factorization and photochemical age parameterization methods indicated that biogenic sources, particularly plant emissions influenced by sunlight, were the primary contributors to the OVOC levels, with biomass burning also being a significant source.

The dataset may benefit the broad research community from the geographical uniqueness of the field site. However, the manuscript should clarify the key analyses of the manuscript to evaluate the scientific merit of the manuscript.

1.  I am not convinced the level of OVOC observed in this manuscript should be considered 'surprising.' surprising is an ill-defiled term for scientific literature. Using the term would be more acceptable if the presented results clearly contrasted with conventional wisdom, which is not the case for the presented dataset. I think OVOCs should be high in the studied area as it is far away from the major emission sources. I would recommend either the authors **drop the term 'surprising,** or make a scientific argument if the observed OVOC levels are 'surprising'

    Response:Accepted. We delete the term 'surprising' in the revised manuscript.

2.  Equation 3 appears too simplistic to account for real-world source distributions. **Many other anthropogenic VOCs produce OVOCs other than benzene**. The study should **discuss how dominant benzene is as an OVOC source** for justification. I understand **isoprene** is the most dominant BVOC on the global scale, but locally, it may not be the case. Without a proper justification, the underlying assumption cannot be prudently established.

    Response: CO, acetylene, and benzene are commonly used as tracers for primary emissions from anthropogenic sources. De Gouw et al. (2005) firstly described the photochemical age-based parameterization method, using acetylene as a tracer for primary anthropogenic emissions. Subsequent studies employing this method for OVOC source apportionment have predominantly used benzene as the tracer for primary anthropogenic emissions (Zhu et al., 2019; Huang et al., 2019; Zhu et al., 2021). In our data analysis, we found that the peak response of benzene was better than that of acetylene, resulting in more accurate quantification. Therefore,

we chose benzene as the tracer for primary anthropogenic emissions in my calculations. Both concentration contribution and the contribution to the rate constant of OH reaction ($k_{OH}$) from benzene to total anthropogenic VOCs suggest benzene is also a major anthropogenic VOC and source of OVOCs.

Isoprene is mainly emitted from plants, although some studies have reported its emission from vehicle exhaust as well. However, in the vicinity of Lulang, where the vegetation cover is dense and anthropogenic emissions such as vehicle exhaust are relatively low, isoprene was selected as a tracer compound for biogenic emissions. Furthermore, based on the diurnal variation of isoprene and the results of PMF source apportionment, it is evident that local isoprene emissions are predominantly biogenic in nature.

3. In the past decade, a substantial progress has made in the atmospheric isoprene oxidation processes, which illustrates the first-generation oxidation product yield (e.g. MVK and MACR) substantially varies **as a function of the NO levels**. The Equation (2) should be reconsidered to reflect the development.

Response:We appreciate the reviewer's comment on the progress in atmospheric isoprene oxidation processes and the potential impact on our Equation (2). We have discussed in more details our method and uncertainties.

The approach adopted in our manuscript assumes that biogenic source of OVOCs and isoprene emission should be proportional under specific conditions, such as solar radiation intensity, temperature and oxidant concentrations and NO concentrations. We have indeed measured similar diel profiles of solar radiation intensity, temperature and oxidant concentrations and NO concentrations from day to day, which supports our assumption. The measured results are reasonable within the monthly measurement period and at a site far from anthropogenic emissions.

Utilizing the measured relationship between typical OVOCs such as MVK and MACR, and their parent VOC isoprene, we estimate the hydroxyl radical exposure [OH]Δt and back-calculate the initial isoprene concentrations. However, we acknowledge that this calculation method has inherent errors and uncertainties. For instance, the calculation primarily considers the chemical conversion of isoprene, neglecting dilution effects. Consequently, the initial concentration of isoprene may be underestimated. Nevertheless, OVOCs and isoprene undergo similar dilution in the atmosphere, suggesting that the OVOC/isoprene ratio is less affected by dilution. In addition. simplified chemical conversion assumption by OH radicals might be another source of uncertainties. The VOC-oxidation reactions initialized by ozone and $NO_3$ radical during nighttime, as well as Cl radicals at daytime, are not negligible. Furthermore, the reaction pathways and OVOC yields from isoprene can vary under different NO levels, further contributes to the uncertainties in determining OVOCs/isoprene ratio and the initial concentration of isoprene. Nevertheless, the oxidation reactions of VOCs initiated by various oxidants under various NO conditions may already be reflected in our measured relationship between MVK and MACR and their parent VOC isoprene. Overall, this method should provide a reasonable deduction result on initial concentration of biogenic VOCs.

Nonetheless, we will expand our discussion on the limitations of our approach in the revised manuscript, addressing potential sources of uncertainty and highlighting the need for further research in this area.

**References:**

de Gouw, J. A., Middlebrook, A. M., Warneke, C., Goldan, P. D., Kuster, W. C., Roberts, J. M., Fehsenfeld, F. C., Worsnop, D. R., Canagaratna, M. R., Pszenny, A. a. P., Keene, W. C., Marchewka, M., Bertman, S. B., and Bates, T. S.: Budget of organic carbon in a polluted atmosphere: Results from the New England Air Quality Study in 2002, Journal of Geophysical Research: Atmospheres, 110, https://doi.org/10.1029/2004JD005623, 2005.

Huang, X. F., Wang, C., Zhu, B., Lin, L. L., and He, L. Y.: Exploration of sources of OVOCs in various atmospheres in southern China, Environ Pollut, 249, 831–842, https://doi.org/10.1016/j.envpol.2019.03.106, 2019.

Zhu, B., Han, Y., Wang, C., Huang, X., Xia, S., Niu, Y., Yin, Z., and He, L.: Understanding primary and secondary sources of ambient oxygenated volatile organic compounds in Shenzhen utilizing photochemical age-based parameterization method, Journal of Environmental Sciences, 75, 105–114, https://doi.org/10.1016/j.jes.2018.03.008, 2019.

Zhu, B., Huang, X.-F., Xia, S.-Y., Lin, L.-L., Cheng, Y., and He, L.-Y.: Biomass-burning emissions could significantly enhance the atmospheric oxidizing capacity in continental air pollution, Environ. Pollut., 285, 117523, https://doi.org/10.1016/j.envpol.2021.117523, 2021.